

# Hillslope-Torrential Hazard Cascades in Tropical Mountains

Maria Isabel Arango-Carmona [1], Paul Voit[1], Marcel Hürlimann[2], Edier Aristizábal[3], and Oliver Korup[1,4]

[1]Institute of Environmental Science and Geography, University of Potsdam, Potsdam, Germany

[2]Departament of Civil and Environmental Engineering UPC-BarcelonaTECH, Barcelona, Spain
[3]Departamento de Geociencias y Medio Ambiente, Universidad Nacional de Colombia, Medellín, Colombia
[4]Institute of Geosciences, University of Potsdam, Potsdam, Germany

*Correspondence to*: Maria Isabel Arango-Carmona (arangocarmona@uni-potsdam.de)

**Abstract**

Torrential hazards refer to the spectrum of water-sediment flows that include debris flows, debris floods, hyperconcentrated flows, and flash floods. These processes often occur in cascading sequences with landslides and have been highly destructive in tropical and humid subtropical mountains, particularly. We compiled a database of 22 cascade events from 2009 to 2024 and analyzed topographic, sediment and the intensity and extremity of both antecedent and triggering rainfall to identify common traits of these events. The results showed that only a few cases were linked to the most extreme rainfall recorded, suggesting that

other controls, such as sediment availability, may be needed for initiating hazard cascades. Clustering analysis revealed that regions with steeper slopes and finer soils experienced hazard cascades even under lower-intensity rainfall, whereas gentler slopes with coarser material required more extreme triggering rainfall. Cascades triggered by both earthquakes and rainfall showed that these triggers can interact across time, such as a rainy season preceding an earthquake or vice versa, or even simultaneously, highlighting the susceptibility of tectonically active tropical regions to cascading hazards. Our findings highlight

the importance of prioritizing hazard assessment and risk reduction strategies in tropical mountains, especially in underreported areas such as Africa.

## 1 Introduction

On December 16, 1999, intense rainfall followed an extraordinarily wet month and triggered shallow landslides and hillslope debris flows across 24 catchments in the mountainous northern Vargas state, Venezuela (García-Martínez & López, 2005). These

flows coalesced with flash floods and hyperconcentrated flows, and evolved into channelized debris flows that surged downstream through steep, narrow canyons, ultimately reaching the urbanized alluvial fans at high speeds. These highly destructive flows carried boulders up to 10 meters in diameter (Larsen et al., 2001), and the sudden onset of the event left many residents trapped in their homes. Within a single day, multiple pulses of debris flows struck the urban areas, killing an estimated 15,000 people, though some estimates suggest the toll could have been as high as 50,000 (Stager, 2009). Over 23,000 houses

were destroyed, and another 65,000 damaged. This remains, to date, the deadliest disaster in Venezuela's history and one of the deadliest in Latin America (García-Martínez & López, 2005).

This disaster is one of the most prominent examples of a type of hazard cascade consisting of a sequence of physically linked processes between hillslope and channels involving mixtures of water and sediment. Reports of similarly destructive events have

become more frequent in recent years, but despite this growing number of case studies, the literature hardly offers more systematic studies. According to EM-DAT, the world's largest disaster database (https://www.emdat.be/), eight out of the ten most fatal disasters from 2000 until 2024 in the categories "flash floods, floods, mudslides, wet avalanches, landslides, and rockfalls" hint in their description at hazard cascades, but they are hard to spot given the lack of a consistent reporting protocol.





Moreover, given the unsteady and non-uniform flow in many torrential and hillslope hazards, the distinction between processes
and their geomorphic evidence is blurred easily, promoting misinterpretation or even confusion of the most dominant or
destructive process. Entries in disaster databases hardly separate single landslides or debris flows from hazard cascades, instead
listing only the process thought to be the most damaging (Cabral et al., 2023; Dowling & Santi, 2014). For example, Legiman et
al. (2023) studied 26 debris-flow events from 1995 to 2022 in Malaysia, noting that at least five involved combinations of debris
flows, debris floods, and mudflows. Kanji et al., (2003) examined 15 debris-flow events in Brazil between 1967 and 2000 and
concluded that the most destructive were those fed by numerous and simultaneous landslides. Jaapar et al., (2023) compiled 15
debris flow events in Malaysia, many of them hazard cascades, which they called "*cascading geological events*", and recognized
as a growing hazard.

Among the disasters recorded in EM-DAT under the categories mentioned before, 273 report more than 100 deaths, with 72%
occurring in the Tropics. A global compilation of fatal debris flows that occurred in 2019 highlighted the two most destructive
events as part of hazard cascades (Prakash et al., 2024), both occurring in Tropical regions, i.e., one triggered by Tropical
Cyclone Idai in Chimanimani, Zimbabwe, claiming 344 deaths, and one triggered by monsoon rainfall in Kerala, India, killing
120 persons. Other countries recently affected by major debris-flow disasters include Zimbabwe, Indonesia, Brazil, India, and
China (Prakash et al., 2024).


The Tropics cover about one-third of the terrestrial land surface, and their environmental conditions, including high sediment
availability, frequent earthquakes, and intense seasonal rainfall events, create optimal conditions for cascading hazards. These
natural conditions overlap with the high vulnerability of tropical countries, in many of which low levels of economic
development contrast with high levels of vulnerability (Sachs, 2001), offering limited options for hazard and risk management. A
more comprehensive analysis of these hazard cascades in tropical mountains is essential to inform risk management and disaster
response strategies. Early warning systems, mitigation measures, and land use regulations might strongly benefit from better
understanding the conditions that lead to hazard cascades.

To address some of the previously mentioned shortcomings and contribute to the knowledge base of hazard cascades consisting
of coupled hillslope and torrential processes in the tropics, our goal is to evaluate how topography characteristics, soil granulom-
etry, sediment connectivity, and the extremeness of antecedent and triggering rainfall influence the initiation and propagation of
these cascades. Additionally, we aim to classify them based on topographic, granulometric, and rainfall-related variables to iden-
tify patterns and distinguish event types.

## 2 Review of hillslope-torrential cascading hazards in tropical mountains

### 2.1 Hillslope – Torrential Cascades

Torrential hazards (Fuchs et al., 2019; Schlögl et al., 2021), also known as catastrophic hydrogeomorphic events (Stoffel et al.,
2013; Wilford et al., 2005), refer to processes that route mixtures of water and sediment down mountain channels. While intense
rainfall is the most frequent trigger, other triggers include volcanic eruptions, earthquakes, and outburst floods caused by glacial,
landslide, or man-made dams (Kaitna et al., 2024a). Torrential hazards include debris flows, debris floods, hyperconcentrated
flows, and flash floods, defined by differing sediment concentrations and flow mechanics (Borga et al., 2014; Iverson, 1997)
(Table 1). Despite some measurable differences in sediment concentration, different stages during a single flow may show either



hydraulic or rheological characteristics (Borga et al., 2014; Costa, 1988; O'Brien & Julien, 1985). Hungr et al. (2014) use the term "debris flow" in a broad sense to encompass a cascading sequence of geomorphic processes that initiate on hillslopes—such as shallow landslides and debris avalanches—and subsequently propagate along drainage channels as debris flows and floods. This integrated definition emphasizes the slope-to-channel sediment transfer and the coupling between hillslope and fluvial

processes at the basin scale, which together define this type of cascading hazard.

**Table 1. Main characteristics of different torrential hazards**

| Type | Subtype | Sediment concentration | Flow characteristics | Source |
|---|---|---|---|---|
| Debris flow | Debris flows | >70% | Rapid, gravity-driven. Peak discharges up to 40x of a major flood. | (Hutchinson, 1988; Nettleton et al., 2005) |
| | Mudflows | >45% clay and silt | Rapid, highly viscous flow. | O'Brien & Julien, (1985b) |
| Debris floods | | > 50% | The movement relies on the traction forces of water. Peak discharges 2-3x of a major flood. | (Borga et al., 2014; Hungr, 2005; Hungr et al., 2001, 2014) |
| Hyper-concentrated flows | | >50%, >15% clay | High yield strength, carries large volumes of sand and some gravel in dynamic suspension. | (Costa, 1988; O'Brien & Julien, 1985; Pierson, 2005) |
| Flash floods | | <4% | Two-phase Newtonian flow. | (Borga et al., 2014; Costa, 1988; Gaume et al., 2009; Marchi et al., 2010) |

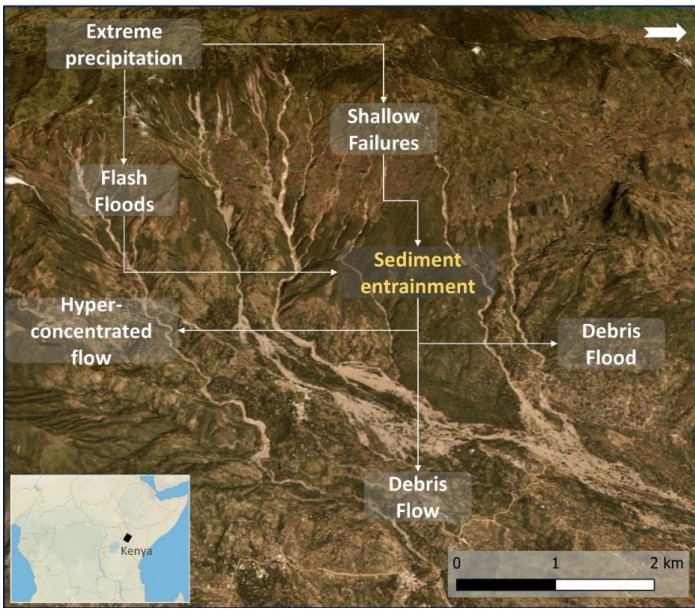

**Figure 1. Example of torrential and hillslope processes and their linkages during HTCs; example shown here is West Pokot, Kenya, in 2020 (Event 15, Table A1). Source: Planet Team, (2017)**



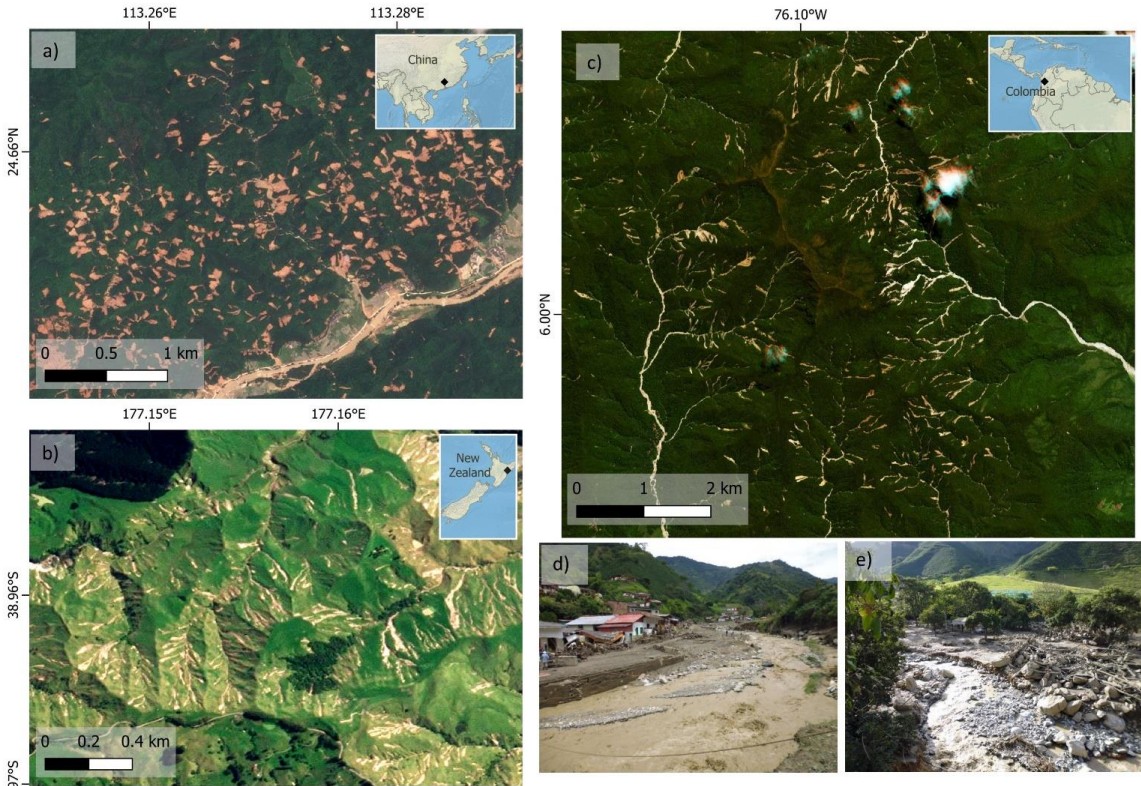

**Figure 2. Geomorphic evidence of Multiple occurrence regional landslide events (MORLE), where sediment from landslides is deposited and retained in the lower slope areas, without being transported through channels: a) Jiangwang, Guandong, China in 2024 and b) Wairoa, New Zealand in 2022. On the contrary, in HTCs like the event in Salgar, Colombia in 2015 (c) (Event 3 in Table A1), sediment entrainment in flooded channels generated rapid channelized debris flows and debris floods (d-e) that destroyed over 60 homes and killed 93 people downstream. Sources: a-c) Planet Team, (2017), d-e) Images courtesy of Jafed Naranjo Guarín.**

Hillslope – torrential cascades (HTCs) are a sequence of interconnected processes that begin on hillslopes as landslides and transfer sediment through various torrential hazards, ranging from debris flows to streamwater floods. More broadly, cascading hazards refer to the phenomenon where an initial event triggers a chain of subsequent processes, which may interact to generate new, potentially more destructive outcomes (Zhang et al., 2023). HTC are mostly initiated by rainfall infiltrating and saturating the soil, reducing its shear strength and triggering shallow landslides (Church & Jakob, 2020; Gregoretti & Fontana, 2008;

Hungr, 2005; Takahashi, 1981) (Figure 1). Sometimes, these landslides are triggered as regional events, where up to tens of thousands of shallow landslides happen in the course of several hours over areas up to 20.000 km$^2$. These events were labelled Multiple Occurrences of Regional Landslide Events (MORLE) by Crozier, (2005). High pore pressure may cause liquefaction and allow the landslide mass to deform like a fluid, creating hillslope debris flows, which reach the channel network if mobile enough (Iverson, 1997; Johnson, 1984). The resulting flow can be a debris flow or debris flood, or it can alternate between the

two, for example when a channelized debris flow outpaces slower-moving flood waves of lesser sediment content; by bed erosion that causes flow bulking; or by an increase in discharge and sediment deposition of the debris flow (Church & Jakob, 2020); and flow blockages caused by woody debris that briefly increase flow momentum. Unlike MORLEs, where the deposits are mostly confined to steep slopes or channels without further interaction with fluvial processes (Figure 2a and b), the sediment





entrainment from debris flows into flooded channels allows HTCs to travel longer distances as heavily sediment-laden flows

(Figure 1 and Figure 2d-g), until reaching torrential fans, i.e. cone-shaped sedimentary landforms at major breaks in slope, for example at mountain-range fronts. Torrential fans are often prime sites for residential development and infrastructure that take advantage of gentle gradients, access to water, and unconsolidated sediment as building materials (Stoffel et al., 2013).

### 2.2 Conditioning and triggering factors of HTCs

Torrential processes depend on three key ingredients: steep slopes, sediment availability, and water (Rickenmann, 2016). The

conditioning and triggering factors for individual torrential processes are well understood (Borga et al., 2014; Kaitna et al., 2024b), but the controls that promote their coupling with each other and with slope processes, and their transformation into a cascade are less clear. These controls likely include the magnitude and duration of the triggering rainfall and the ratio of sediment and water to generate and maintain highly mobile flows. Also, positive feedbacks might amplify the effects of each process, causing the terrain to cater to both sediment and fluvial connectivity to support the process cascade. The following

sections describe the three key conditioning and triggering factors of HTCs: rainfall, topography, and sediment availability and connectivity.

### 2.2.1 Rainfall

Intense rainstorms trigger HTCs by providing abundant water in a short period to saturate soils or exceed their infiltration capacity, causing excess surface runoff, landslides, and channel erosion. Intense rainfall episodes are created or enhanced by

topography in three possible ways: i) by promoting the vertical motion of air masses, causing condensation (Kaitna et al., 2024b); ii) by developing low-level clouds by the blockage of air masses that feed from upper clouds and create a seeder-feeder mechanism (Fernández-González et al., 2015); and iii) by the rapid rise of potentially unstable air masses above its level of free convection (Caracena et al., 1979; Galewsky et al., 2006).

Tropical rainfall is strongly influenced by the Intertropical Convergence Zone (ITCZ), which drives high-intensity rainfall events as part of monsoons and cyclones. The local convergence of air masses along monsoon shear lines, the presence of deep convection zones, and the interaction of factors like the Coriolis force, zonal thermal low-pressure troughs, and the Madden-Julian Oscillation (MJO) all contribute to intense rainstorms in the Tropics (Callaghan & Bonell, 2005; Madden & Julian, 1971). Although highly variable, the frequency of high-intensity rainfall events (>25 mm/h) is greater in the Tropics than in any other

region of the world (Encalada et al., 2019); convective and cyclonic storms significantly contribute to these extreme events (Syvitski et al., 2014).

Antecedent soil saturation and snowmelt can significantly increase terrain susceptibility to torrential hazards by enhancing the likelihood of failure (Prenner et al., 2019). When soils are already moist or saturated due to previous rainfall events, their

capacity to absorb additional precipitation decreases markedly, increasing the probability of surface runoff and shallow landsliding. In tropical mountain regions, where precipitation can be both frequent and intense, short intervals between storms often do not allow sufficient time for soils to drain, leading to a cumulative saturation effect. This preconditioning plays a critical role in triggering hillslope failures during subsequent storms, even if rainfall intensities are moderate.



### 2.2.2 Topography

A common approach to distinguish catchments prone to torrential hazards relies on their morphometric characteristics. Basins that generate debris flows are generally steeper and smaller than those where debris floods or floods occur and have less developed channel networks that generate very high peak discharges upon intense rainfall (de Haas et al., 2024). Catchments dominated by debris flows have values of morphometric variables like catchment area, slope, and drainage length that lie between those dominated by fluvial floods and flash floods, and those dominated by gravitational processes. Yet, catchments

may respond by different torrential processes to the same level of rainfall intensity (Borga et al., 2014; Wilford et al., 2004). Moreover, several studies indicate that morphometric characteristics or thresholds derived from higher latitudes may not be applicable to the Tropics: morphometric parameters of catchments prone to torrential hazards in tropical and temperate regions (Bertrand et al., 2013; Jackson et al., 1987; Melton, 1957; Welsh & Davies, 2011; Wilford et al., 2004) may differ in their Melton Index (Melton, 1957), average fan slope, or debris flow travel distance (Arango et al., 2021; Dias et al., 2022; Lin &

Jeng, 2000). Adapting such indices to differing environments is thus problematic, because torrential hazards are likely conditioned and triggered by a combination of environmental, geologic, and climatic factors rather than topographic features only (de Haas et al., 2024).

### 2.2.3 Sediment availability and connectivity

The availability and type of sediment determine how a mountainous area responds to heavy rainfall, and catchments can be

classified as transport-limited or supply-limited (Bovis & Jakob, 1999). Transport-limited watersheds have high amounts of sediment and recharge rates, such that torrential events depend only on sufficient rainfall. Supply-limited watersheds are often bedrock-dominated and tend to have less frequent, but more extreme (i.e. with higher sediment volumes), torrential events compared to transport-limited basins (Bovis & Jakob, 1999; de Haas et al., 2024).

Sediment connectivity refers to the downstream mobility of loose materials within a catchment and estimates the potential for particles to move towards the catchment outlet (Hooke, 2003). A high sediment connectivity can enhance the connection between hillslopes and torrential processes. For example, during Hurricane Maria in Puerto Rico in 2017, Bessette-Kirton et al., (2020) found that coalescing landslides and those entering channels had increased mobility, likely due to sediment entrainment into floodwater and aggregation from various landslide sources. The study noted that even though most landslides were single,

coalescing landslides and those entering channels make up most of the affected area. Controls on sediment connectivity include terrain morphology; barriers like stream blockages or dams; catchment shape; drainage density; slope; and surface roughness (Cavalli et al., 2013); sediment shape, volume, and particle size; and land cover (Bracken & Croke, 2007).

### 3 Methodology and Data

We followed a methodology comprising three steps to evaluate the influence of topographic characteristics, soil granulometry,

and triggering rainfall in the initiation and propagation of HTCs. First, we compiled a database of documented HTCs, delineating their Area of Interest (AOI). Then we collected data on rainfall, topography, and sediment characteristics for each event in our database. Finally, we applied a clustering analysis to identify patterns and similarities among study regions based on such characteristics.



### 3.1 Database compilation

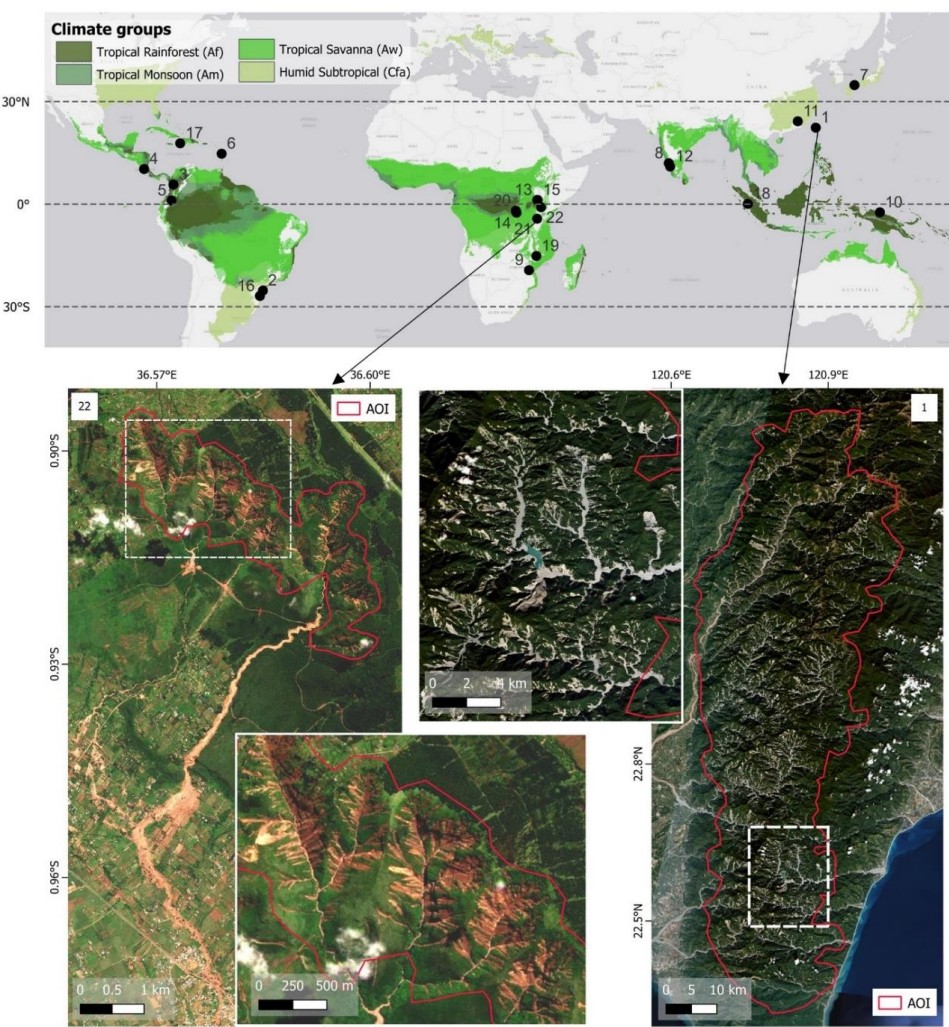

**Figure 3. Locations of the 22 HTCs compiled in our database in tropical and humid subtropical regions; event numbers correspond to those in Table A1. The largest event was in Taiwan in 2009 (Event 1 in overview map and in Table A1) with an AOI of 3,827 km², and the smallest in Kenya in 2024 (Event 22) with an area of 6 km². Satellite images: Planet Team, (2017).**

We compiled a database of 22 HTCs that occurred between 2009 and 2024 in tropical or humid subtropical mountains as defined by Peel et al., (2007) (Figure 3). We consulted multiple sources, including disaster databases such as Desinventar (Desinventar Sendai, 2024), the Landslide Blog (Petley, 2024), and the Disaster Charter (International Charter Space & Major Disasters, 2020), local reports, and optical satellite imagery from Planet (Planet Team, 2017) at a 3-m resolution. We cross-checked all events using scientific literature and technical reports to compile detailed information, including the number of fatalities, affected area, and number of landslides. The trigger for each event—classified as rainfall or rainfall and earthquake—was adapted from these sources. For rainfall-triggered events, we gathered data on rainfall intensity and reported duration. Where stated, we included information about the dominant lithology of the affected areas (Table A1).





We have delineated an Area of Interest (AOI) for each event (red polygons in Figure 3). With a focus on the source areas of
cascading hazards instead of their full extent, we excluded areas affected solely by sediment runoff or deposition. Thus, we
manually delineated the areas with visible entrainment of hillslope debris flows into channels as AOIs from Planet satellite
imagery. We excluded areas without a clear coupling between slopes and channels or those dominated solely by channel or fan
processes.

### 3.2 Terrain, rainfall and sediment data analysis

### 3.2.1 Topography and sediment connectivity

We extracted local slope and drainage area from 12.5-m digital elevation models (DEMs) derived from ALOS PALSAR (ASF
DAAC, 2015). Soil granulometry data (percentages of clay, silt, and sand) were obtained from the SoilGrids database (Poggio et
al., 2021) at a 250-m resolution, averaged for a depth of 2 m, and rescaled to match the DEM resolution. We followed Cavalli et
al., (2013) to estimate sediment connectivity from the DEM. The sediment connectivity (IC) is dimensionless index that
estimates the linkage between upslope and downslope components of connectivity. The upslope component depends on the slope
gradient and contributing area, while the downslope component is influenced by flow path length and slope at a given location.
Sediment connectivity can be assessed relative to specific target locations, such as catchment outlets.

We computed sediment connectivity at two levels: "IC Slopes" refers to the connectivity measured in the hillslopes using all
pixels located in the drainage network as target locations, and "IC Drainages", which refers to the connectivity of the pixels
corresponding to the drainage network, measured using the basin outlets as the targets. We used the SedInConnect tool (Crema
& Cavalli, 2018) to compute the IC values of the two levels.

### 3.2.2 Rainfall analysis

We extracted CHIRPS gridded daily rainfall data (0.05° resolution, i.e. ~5.5 km at the equator) from 1981 to 2024 to estimate the
rainfall intensity and extremeness in HTCs across spatial and temporal scales.

We began by analyzing the relationship between the intensity of triggering and antecedent rainfall. To do this, we aggregated
daily rainfall records, provided in millimeters per day (mm/day), defining triggering rainfall as either the rainfall accumulated on
the event day alone or the total rainfall accumulated over the event day and the two preceding days. Antecedent rainfall is
defined as the total rainfall accumulated over the 90 days preceding the event.

To assess how extreme these rainfall events are, we computed the cross-scale Weather Extremity Index (xWEI) (Voit &
Heistermann, 2022) which defines extremeness (EtA) as the product of rarity (return periods) and affected area extent (Müller &
Kaspar, (2014). A high xWEI indicates extreme rainfall over different durations and a large area (Voit & Heistermann, 2022).
For this, we aggregated six different durations (1, 3, 6, 30, 60, and 90 days) and analyzed each of them by extracting yearly
maxima for each CHIRPS cell and then fitting a duration-dependent Generalized Extreme Value (dGEV) distribution
(Koutsoyiannis et al., 1998), which accounts for dependencies across durations. For example, an extreme daily rainfall is likely
also contained in an extreme three-daily rainfall. To assess which rainfall duration may have influenced the occurrence of the
HTC most, we calculated the extremeness (EtA) for each duration separately (Figure 4a and b). We used a day search window of
3 times the duration (e.g., 9 days for a 3-day duration) before the event and aggregated the rainfall within this search window by
a moving window for each duration. We then computed the extremeness of the entire rainfall event following Müller & Kaspar,





(2014) (Figure 4c). We also computed daily xWEI values for the full CHIRPS dataset. The resulting time series helped to identify the antecedent rainfall leading up to HTCs and comparison with similar past events. To compare the EtA values across different regions, we normalized these values to the maximum EtA found in the whole CHIRPS dataset for each AOI.


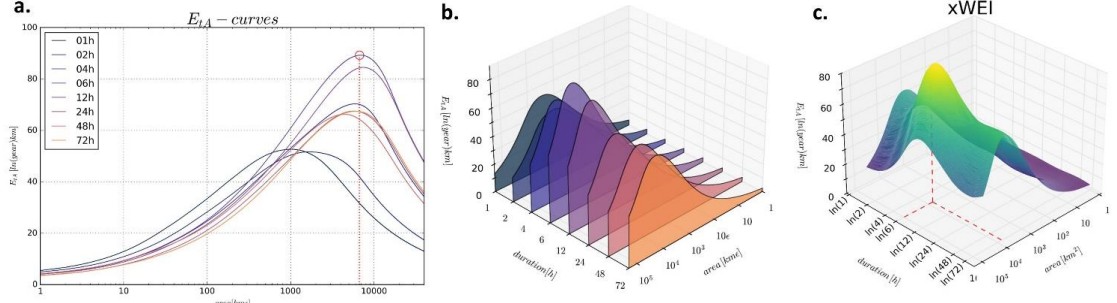

**Figure 4. a) $E_{tA}$ curves of rainfall extremity for different durations and areas. b) $E_{tA}$ curves aligned per duration, normalized with respect to the highest $E_{tA}$ per duration and area. c) values from (b) interpolated on a grid of logarithmic values of durations and area. The xWEI is the area under the curve and describes the extremity of the rainfall across duration and extension.**

### 3.3 Clustering analysis

We performed a *k*-means clustering analysis to identify patterns and similarities among study regions, using 19 topographic, granulometric, and rainfall variables shown in Table 2. To avoid relying solely on mean values of the variables per AOI, we

clustered using CHIRPS pixels as the sample size (~5.5 km resolution). This approach resulted in a variable number of samples per study area, ranging from 2 to 256 pixels, depending on the spatial extent of the different AOIs.

Before clustering, we applied Principal Component Analysis (PCA), which employs singular value decomposition (SVD) to reduce the number of dimensions of the dataset while retaining its variance. We used the prcomp() function from the stats package in R. Before running PCA, we standardized the dataset to ensure all variables had equal weight in the analysis. We selected

the first five principal components (PCs), which together explained 87% of the total variance, as input for the clustering analysis.

To determine the optimal number of clusters, we evaluated the Silhouette score, a metric that assesses how well each data point fits within its assigned cluster relative to other clusters. For clustering, we used *k*-means with Euclidean distance, using the kmeans() function from the stats package in R. After defining the clusters, we examined the distribution of key variables within each group and analyzed the PCA loadings that show how original variables contribute to each principal component. This al-

lowed us to identify the commonalities and differences across groups of documented events.

**Table 2. Variables used in the *k*-means cluster analysis, divided into the four main ingredients for HTC generation: topography, soil composition, and rainfall, expressed as intensity and extremity.**

| Topography | Soil granulometry (% of soil mass in soil depth of 2m) | Rainfall intensity (mm /day) | Rainfall extremity (% of max) |
|---|---|---|---|
| Slope (degrees) | Mean clay content | 1 day | xWEI |





| | | | $E_{tA}$ 1 day |
| | | 3 days | |
| | | | $E_{tA}$ 3 days |
| | | 6 days | |
| IC Slopes | Mean silt content | | $E_{tA}$ 6 days |
| | | 30 days | |
| | | | $E_{tA}$ 30 days |
| | | 60 days | |
| IC Drainages | Mean sand content | | $E_{tA}$ 60 days |
| | | 90 days | |
| | | | $E_{tA}$ 90 days |

## 4 Results

### 4.1 General analysis of database

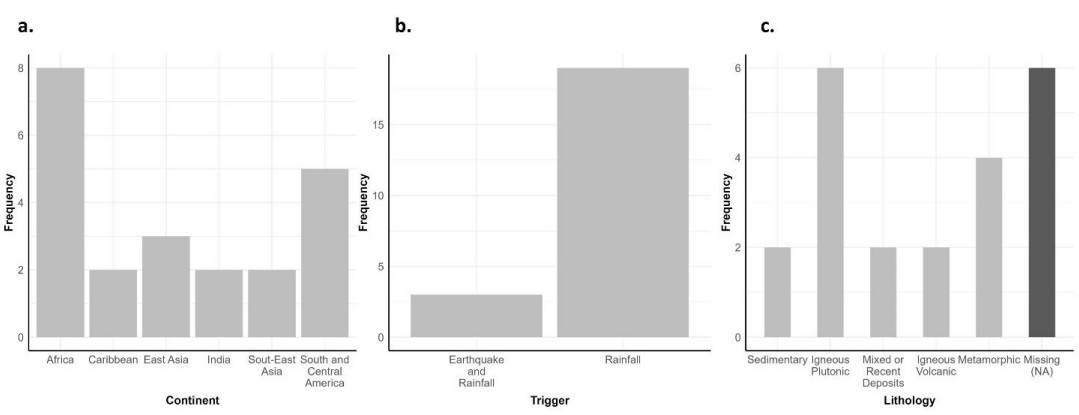

Figure 5. Locations, triggers, and dominant lithology in areas affected by a global sample of 22 HTCs.

We compiled 22 HTC events across all Tropical world regions (Figure 5a). We refer to each HTC in terms of their AOIs, followed by their number in Table A1 in parentheses. The median size of all the AOIs that we mapped is 33 km², although their sizes vary greatly. The smallest AOI is Kenya (22); that was triggered by an intense, localized rainstorm, and had an area of 6 km², though the resulting debris flow travelled some 10 km (UNITAR & UNOSAT, 2024). The largest AOI is Taiwan (1), triggered by Typhoon Morakot with HTCs occurring in an area of >3,800 km² (Figure 3). According to Lin et al., (2011), the

total affected area, including deposition zones not considered in our AOI, was 7,811 km².

We found that 19 of the 22 documented HTC were triggered by rainfall, and five of these were associated with tropical cyclones. The remaining three events were triggered by the combined effects of earthquakes and rainfall (Figure 5b), although in varying sequence. The 2016 event in Bijagua, Costa Rica (4) was attributed to Hurricane Otto, but the area was shaken by a $M_w$ 5.4

earthquake four months before (Quesada-Román et al., 2019). The 2021 event in Pic Macaya, Haiti (17), was described as co-seismic landslides triggered by the 7.2 $M_w$ Nippes earthquake by Martinez et al., (2021) and Zhao et al., (2022). However, both sources report that Tropical Cyclone Grace passed through the region two days later. While noting the cyclone, they remain inconclusive about the trigger because of a lack of satellite imagery between the earthquake and subsequent rainfall. Similarly,



the 2022 Mont Talakmau event in Indonesia (18) is classified as co-seismic landslides triggered by the 6.2 $M_w$ Sumatra

earthquake (Fang et al., 2024), with no explicit links to rainfall. However, some rain had occurred in the days prior and on the day of the event. In both the Haiti and Indonesia cases, the high mobility of the landslides and their transition into channelized debris flows that traveled over 10 km (Figure 6) indicate some contribution of rainfall. It is likely that both seismic shaking and rainfall played a role in triggering and sustaining the HTC.

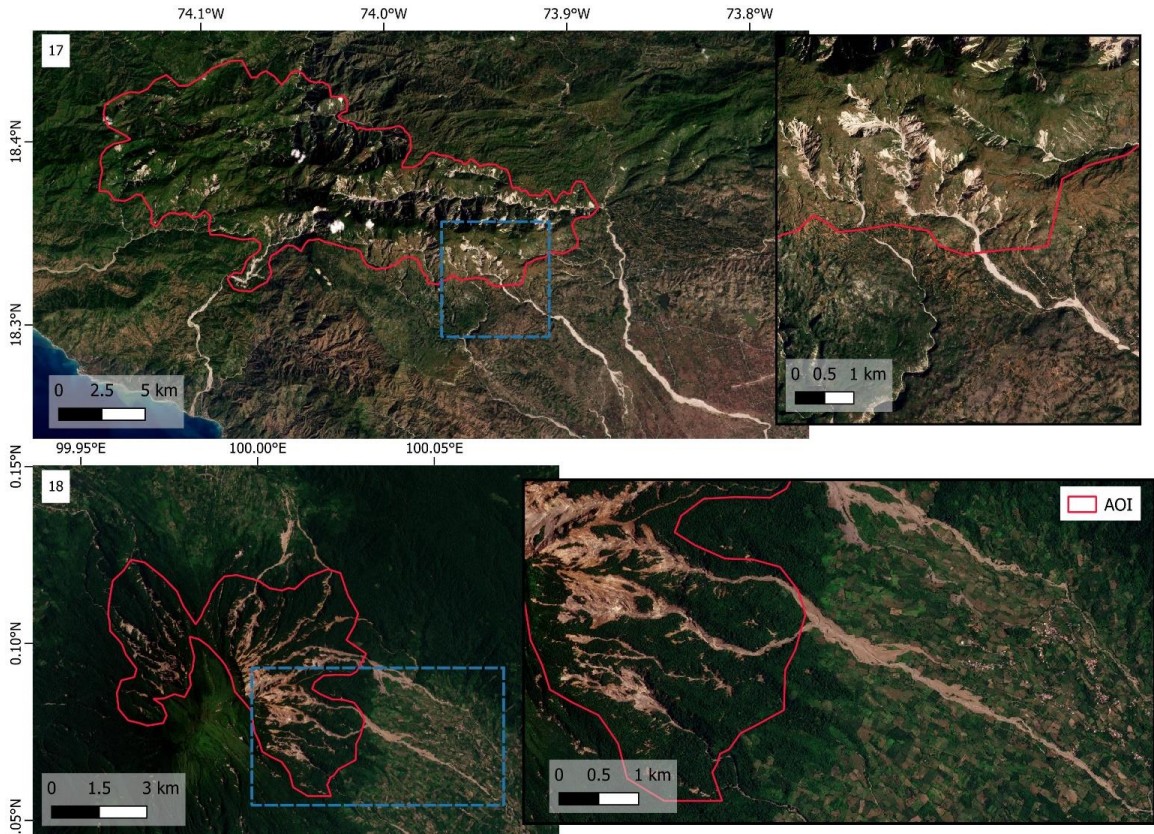


**Figure 6. Two HTCs reported to be triggered by earthquakes. The long runouts indicate highly mobile flows, suggesting the combined action of the seismic shaking with nearly coincident rainstorms. Top: overview (left) and zoom-in (right) of Pic Macaya, Haiti in 2021 (17), and bottom: Mt. Talakmau in Indonesia in 2022 (18). Blue-dashed rectangles indicate zoom-in areas. Satellite images: Planet Team, (2017).**

The dominant lithologies for a dozen of the HTCs were grouped into general categories (Figure 5c), and the results show that plutonic rocks were the most common, followed by metamorphic rocks. For Japan (7), Rodrigues Neto et al. (2023) highlighted that granitic and granodioritic rocks in the area were more susceptible to landslides than volcanic rocks, because the former were

weathered into permeable and brittle sand. Similarly, for Colombia (5), García-Delgado et al., (2019) reported that highly tectonically weathered granitic rocks were highly susceptible to landslides, while sedimentary rocks promoted slope failures through water infiltration along weathered shear fractures.



Landslide density varied across events, with ten events having mapped landslides of different extents. Reported landslide
densities thus vary widely, ranging from 0.08 landslides/km² in India (8), where over 300 large landslides were scattered across
the 4,100-km² Kodagu rural district  (Meena et al., 2021), to 35 landslides/km² in Costa Rica (4), where more than 900 small,
shallow landslides happened over 27 km² on the slopes of Miravalles volcano (Quesada-Román et al., 2019).

### 4.2  Topography, sediment and rainfall characteristics

### 4.2.1 Topography and sediment characteristics

There is substantial variability in the terrain and sediment characteristics across the AOIs, partly because of the large differences
in their size (Figure 7). The mean slope across the regions had moderate spreads, varying from 14.7° to 37.1° with a median of
26.3°. While sediment connectivity of streams (IC Streams) shows only moderate spread, the IC of hillslopes (IC Slopes) varies
more widely, with a high spatial variability across each AOI. No single granulometric composition dominates across the AOI, as
clay, silt, and sand content vary greatly.


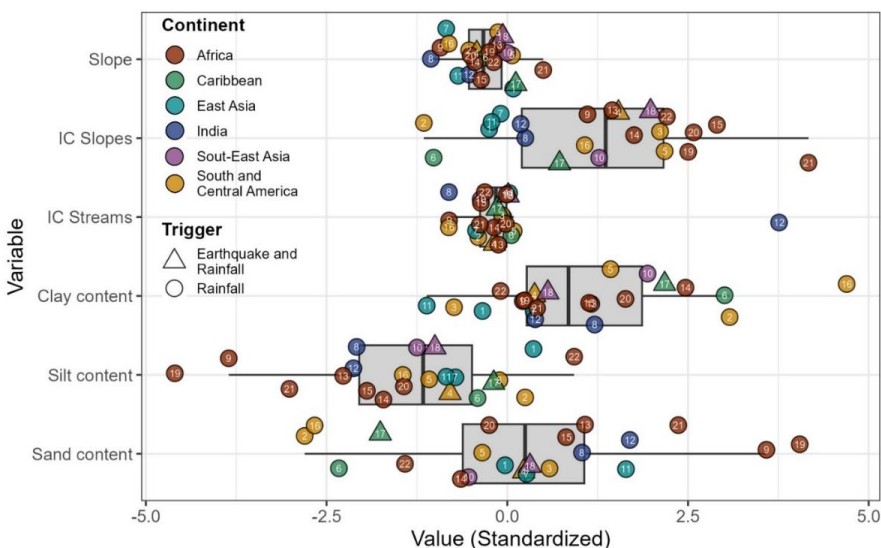

**Figure 7. Distribution of terrain and sediment granulometry variables of 22 HTCs, standardized to the global mean across all study
areas: zero refers to the mean, and units are standard deviations. Circles and triangles are mean values for each study area, and
numbers correspond to the event (see Table A1). Box-and-whisker plots show medians as thick vertical lines; boxes span the lower and
upper quartiles, and whiskers extend over 1.5 times this interquartile range.**

### 4.2.2 Rainfall characteristics

Rainfall characteristics are analyzed in terms of intensity (Figure 8) and extremity (Figure 9 and Figure 10). Figure 8 shows the
relationship between 1-day and 3-day with the 90-day rainfall accumulation for each AOI. The triggering rainfall intensities
varied between 10 and 100 mm, and 90-day totals from 700 to 1300 mm. The lowest triggering and antecedent rainfall for a HTC
was associated with an event in Brazil (16) with only 11 mm on the event day, and 370 mm in the 90 days before. The data show
a weak linear relationship between triggering and antecedent rainfall (Figure 8, $r$=0.31 for 1-day rainfall; $r$=0.7 for 3-day
rainfall). AOIs in East Asia and India stand out from the rest of the regions due to the intensity of their antecedent and triggering





rainfall, especially at the 3-day duration, which is notably higher than in other regions. This includes Taiwan (1) triggered by

Typhoon Morakot, India (8 and 12), triggered by intense monsoon rainfalls, with the latter having had a 90-day accumulated rainfall of 2717 mm, being the highest of the database: and Japan (7), caused by Typhoon no.7, which dropped 228 mm in a single day and 355 mm in 3 days, being the most intense triggering rainfall recorded in the database.

No events were reported following a combination of high antecedent rainfall and low triggering rainfall, or by intense short-term

rainfall combined with low antecedent rainfall.

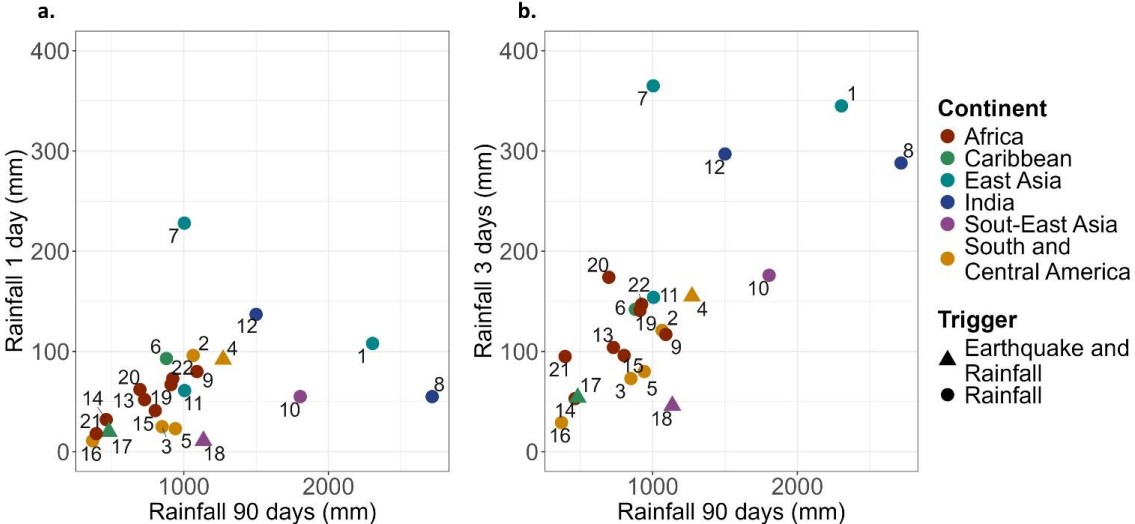

**Figure 8. Relationship between 90-day antecedent rainfall and triggering rainfall of 1 day (a), and 3 days (b) before the event based on CHIRPS rainfall data for each AOI. The numbers correspond to AOI numbers in Table A1.**

In addition, Figure 9 summarises the extremity of rainfall events, analyzed using the xWEI of each event and the corresponding $E_{tA}$ values for various rainfall durationsFigure 9. Only two of the 22 HTCs were triggered by the most extreme recorded rainfall event ever measured in their AOI, i.e. the Democratic Republic of the Congo (20), and Japan (7), both of them had high

extremeness in their 1-day and 3-day rainfall durations. Only four events were triggered by events with xWEI values at more than half of the most extreme recorded; and half were below 10% of the most extreme one. Overall, the extremeness of the rainfall that triggered the events in our database is greater in the long term (30, 60, and 90 days) than in the short term (1, 3, and 6 days). Among all the durations, the extremeness in the 1-day duration is the lowest, and the 90-day is the highest. Events triggered by earthquakes and rainfall remain in the region of low medium to low rainfall accumulation in both axes.


Figure 10 shows the time series of xWEI and rainfall intensity ($E_{tA}$) across different durations for four HTCs triggeded by extreme rainfall events. The events in Japan (7) and DRC (20) coincided with the highest xWEI on record. However, three other events—Indonesia (10), Brazil (2) and Kenya (22)— occurred within three days before the peak of another very extreme rainfall. These events were triggered before the rainfall reached its maximum intensity, indicated by their lower xWEI values on the day

of occurrence. When comparing the xWEI values of all other HTC with their time series, only three fell within the top 100 most extreme events.



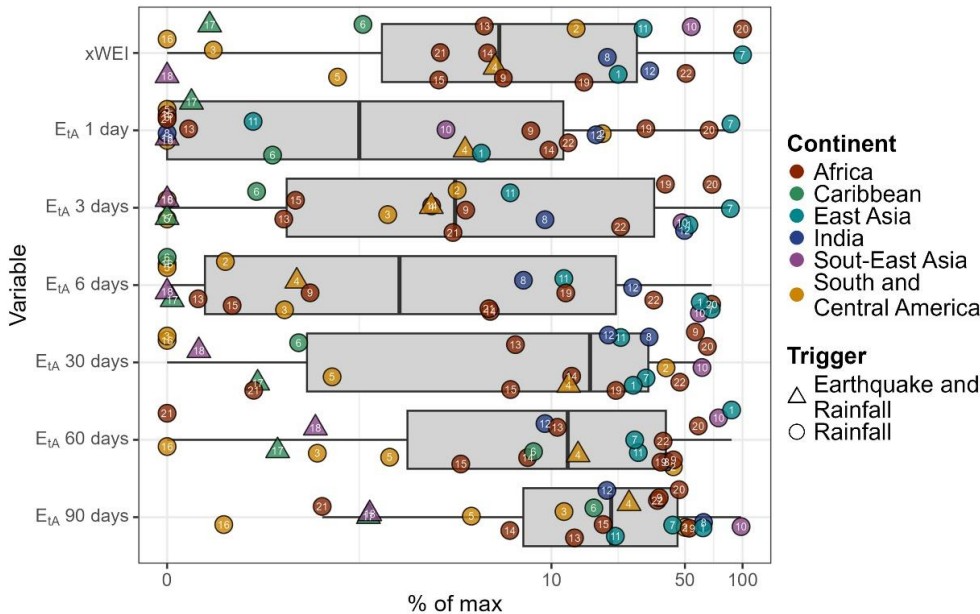

**Figure 9. Distribution of the xWEI rainfall extremeness index for rainfall of 1- to 90-day duration and various spatial extents, along with the corresponding extremity ($E_{ta}$). All values are normalized to the maximum recorded for each AOI, such that a value of 100% is the most intense rainfall observed in the AOI between 2000 and 2024. See Fig. 7 for box-plot explanations and Table A1 for numbers.**

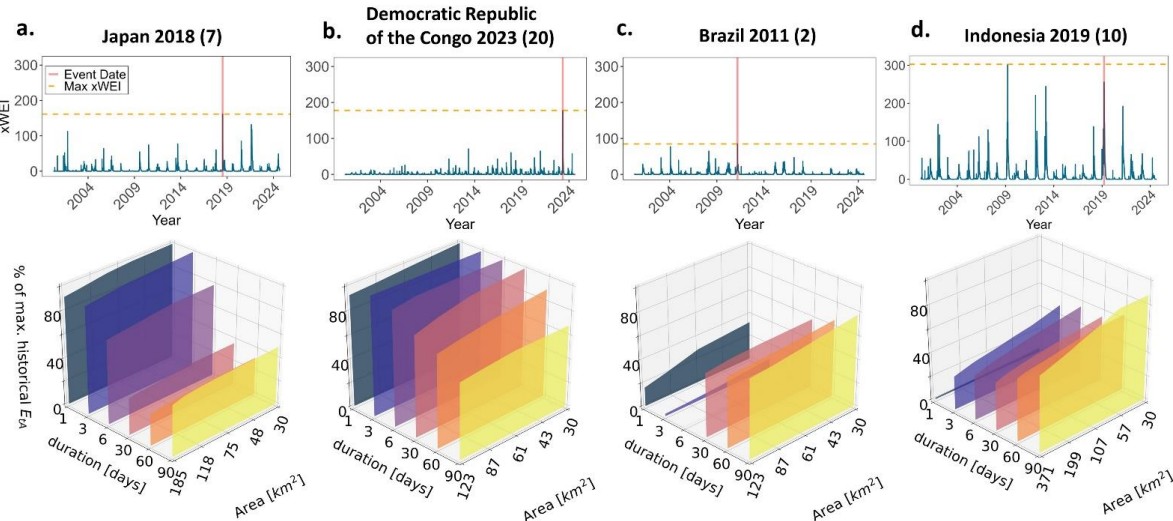

**Figure 10. Summary of rainfall time series and extremes of four selected HTCs. (a–d) Upper plots show time series of xWEI between 2000 and 2024 with the day of the HTC highlighted and compared with the highest recorded xWEI; lower plots show triggering rainfall intensity ($E_{ta}$) of the HTC across different rainfall durations and extensions. The xWEI values vary in scale across AOIs because of their size and the fitted GEV distributions. In (a) and (b), short-term rainfall (1–6 days) contributed most to event extremeness, whereas in (c) and (d), long-term rainfall contributed more.**





The extremity index xWEI allows to also underlines the localized nature of the rainfall that triggered the HTCs. In all the AOIs, the extremity was higher at the local scale than in the regional one. Figure 11 shows three examples of this: In Taiwan (1), the most extreme rainfall occurred over 3-, 6-, and 60-day durations, with peak intensities concentrated over 118 km², or only 2% of the total affected area. In India (12), the highest extremity was observed over 3 days, with its peak rainfall affecting 25% of the area. In contrast, in Malawi (19), the most extreme rainfall was over 90 days, with rainfall distributed more evenly across the

AOI. However, even in this case, short-term extremes of 1 and 3 days were more localized and detected using CHIRPS data despite its coarse resolution.

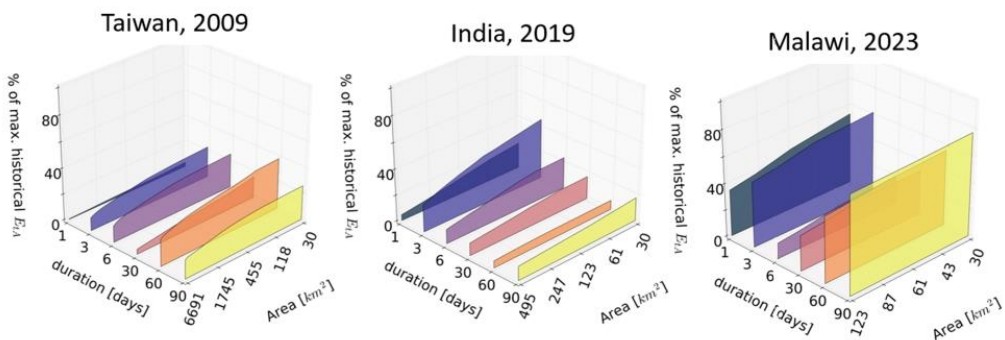

**Figure 11. Triggering rainfall intensity ($E_{ta}$) of three HTCs across different rainfall durations highlights higher extremity values across smaller areas.**

### 4.2.3 Relationships between topography, sediment and rainfall characteristics

    Figure 12 shows the correlation between topographic, sediment and rainfall values. The rainfall was assessed in terms of both

intensity and extremity) at the CHIRPS pixel scale (~ 5.5 km²), where each pixel had different topographic and rainfall values, but all pixels from the same AOI had the same $E_{tA}$ and xWEI valuesFigure 12. We stress the lack of correlation between the sediment connectivity (IC) of slopes and the rainfall extremity (xWEI) with all other variables. The IC of channels and clay content are negatively correlated with both rainfall intensity and extremity of specific rainfall durations, indicating that AOIs with clay-rich soils and higher sediment connectivity in streams are triggered by less intense and less extreme rainfall.

### 325  4.3 Cluster Analysis

    The silhouette scores were used to determine the optimal number of clusters. Although the analysis suggested an optimal number of seven clusters, we chose to limit the data to three groups to prevent overfitting and given the small sample size. Clusters were defined based on the first five principal components, which together explain 87% of the total variance.

Figure 13a show a scatterplot of all analyzed pixels in the space defined by the first two principal components (PC1 and PC2), which account for 48% and 19% of the variance, respectively. Each point represents a pixel, and the ellipses outline the extent of each cluster. Figure 13b shows the variable loadings of each PC, where PC1 is primarily influenced by variables of rainfall extremity across different durations, and PC2 is associated with rainfall intensity and terrain characteristics such as soil granulometry and slope.






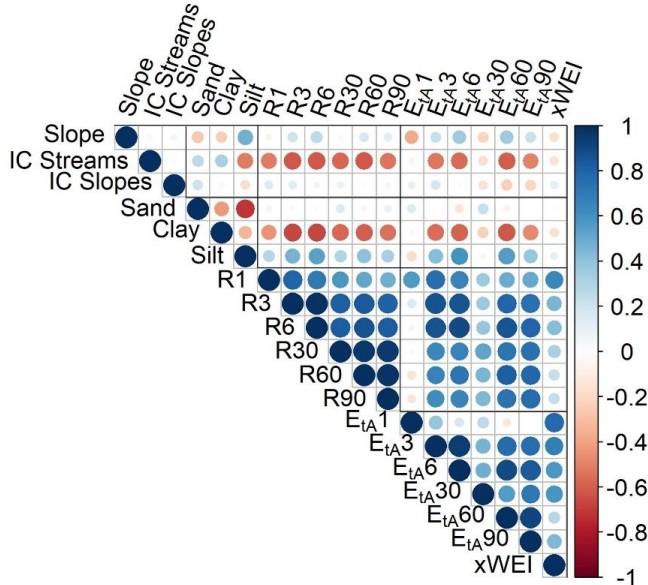

**Figure 12. Correlation matrix for terrain and rainfall variables measured at the scale of the CHIRPS pixels. The darker lines divide the variables into groups: topography, sediment granulometry, rainfall intensity, and rainfall extremity. Ri variables refer to the accumulated rainfall the corresponding days, i, before the HTC. $E_{tA}i$ refers to the percentage of the extremity of the corresponding rainfall duration in i days, compared to the maximum recorded.**

Figure 14 shows the distribution of all 19 standardized variables across the three clusters, enabling a direct comparison of how variables differ between them. For instance, Cluster I includes nine AOIs, characterized by gentle slopes, sandy and silty soils,

and HTCs triggered by intense rainfall, particularly for a 1-day duration. Lithologies are predominantly granitic and metamorphic. Cluster II includes ten AOIs, including the three events triggered by both rainfall and earthquakes. These AOIs have intermediate to steep slopes, high stream sediment connectivity, and fine-grained soils (silt and clay). These HTCs were mostly triggered by rainfall events of lower intensity and extremity, especially over short durations. Their lithologies include sedimentary rocks, volcanic deposits, gneiss, and granite. Cluster III contains two AOIs with exceptionally steep slopes, high silt

content, and HTCs triggered by prolonged and extreme rainfall over 60–90 days.

## 5 Discussion

In this section we aim to explore the key findings from our analysis of 22 hillslope-torrential cascades (HTCs) in tropical regions between 2009 and 2023. We focus on understanding the similarities and differences in the rainfall conditions that trigger these events, alongside the susceptibility factors like topography, soil granulometry, and sediment connectivity.


None of the analysed HTC events was triggered solely by intense short- or long-term rainfall; instead, a combination of both played a role. However, rainfall extremeness was greater over longer durations (30, 60, and 90 days) than in the short term. While short-term rainfall contributed to the events, HTCs often followed particularly extreme rainy seasons. This long-term rainfall accumulation may contribute to reducing soil stability, increasing the likelihood, number, and mobility of hillslope debris

flows triggered by intense (but not necessarily extreme) rainfall events, and increasing runoff due to soil saturation. This is





supported by other studies about the role of antecedent rainfall conditions for phenomena like landslides and torrential flows flows (Guzzetti et al., 2008; Kim et al., 2021; Prenner et al., 2019).

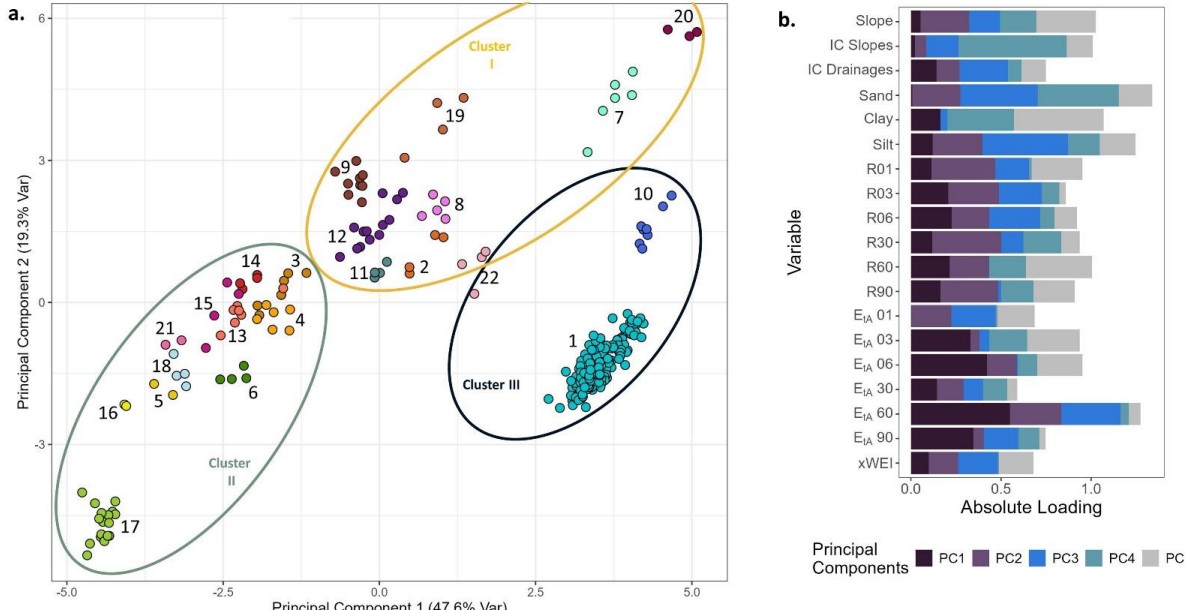

**Figure 13. a) Distribution HTCs based on their first two principal components (PC1 and PC2), with points colour-coded by Area of Interest (AOI); and numbered according to Table A1. Ellipses outline the extent of each of the three *k*-means clusters. b) Variable loadings of the five principal components used in the clustering analysis, indicating the contribution of each standardized variable to the PCs.**

To explore the unusually low rainfall intensities that appeared to trigger some events, we compared CHIRPS rainfall estimates with reported rainfall accumulations in event reports or descriptions. These reported triggering rainfall ranged from 1 mm/h to 43 mm/h. Yet, comparing these records is difficult as some rely on rain gauge measurements taken far from the event sites, while others use radar-based estimates or satellite imagery. When comparing these reports with CHIRPS estimates, we found large differences, especially for short-term rainfall. For example, Velásquez et al., (2020) reported that the 2015 Colombia event (3)

was triggered by a localized convective storm of 180 mm of rain overnight, based on radar data. However, CHIRPS estimated 25 mm on the event day. Similarly, for the 2020 Brazil event (16), CHIRPS estimated 11 mm on the event day, while Michel et al., (2021) reported 124 mm in a single day based on local rain gauge data. These inconsistencies were evident throughout the database, with absolute differences between reported and actual values decreasing as the accumulation period increased. This finding aligns with that by Urrea et al., (2016), who compared CHIRPS rainfall estimates with gauge measurements in Colombia,

indicating that the coarse resolution of CHIRPS tends to underestimate short-term rainfall variability, smoothing localized extremes while performing better at monthly and annual scales, as errors decrease.



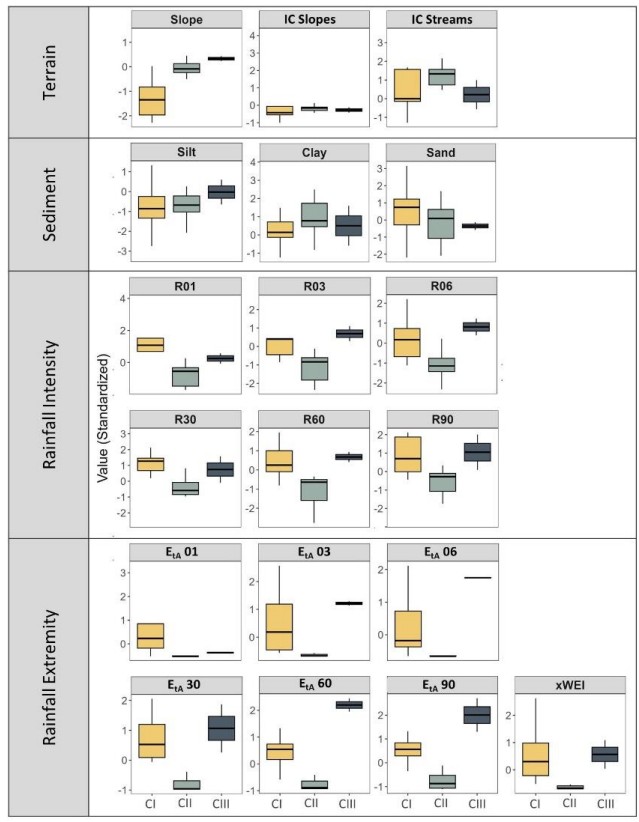

**Figure 14. Distribution of standardized values for terrain, sediment, and rainfall variables across the three identified clusters (CI, CII, CIII). Boxplot colors and cluster labels match those shown in Figure 13a.**

Only two HTCs in our database were triggered by the most extreme rainfall recorded in the AOI across all durations from 1 to 90 days; three others occurred just before the peak of very extreme events. Still, most (i.e. 15) HTCs were associated with rainfall that had an extremity index <25%. This finding suggests that these events occurred in supply-limited watersheds, and thus not as a sole consequence of extreme rainfall, but also of sediment availability. The frequency of torrential events in supply-limited basins depends on the specific sediment recharge rates, producing less frequent but more extreme events than transport-limited

basins do, where unlimited sediment supply means torrential events are triggered more by critical rainfall (Jakob & Hungr, 2005). Moreover, the Tropics have some of the highest rates of biogeochemical weathering (Syvitski et al., 2014). Sustained precipitation and humidity have led to the development of weathered soil profiles tens of meters thick with distinct material contrasts between colluvial layers, saprolitic and residual soils, and bedrock. These contrasts favor slope failure with substantial amounts of poorly consolidated material (Amarasinghe et al., 2023; ). In tectonically active tropical mountains such as parts of

the Andes, the Ethiopian Highlands, or the New Guinea Highlands, physical weathering is also important, while volcanic activity and seismicity provide mobile sediment. These high sediment production rates in the Tropics could mean that HTCs can have shorter recurrence periods than in temperate areas.



In three cases, both earthquakes and rainfall coincided as possible triggers. One case involved intense rainfall triggering a HTC
in terrain that had been weakened by a previous earthquake, as in Costa Rica (4) (Quesada-Román et al., 2019). In Indonesia
(18), a rainy season was followed by an earthquake that triggered the HTC. A third case features the reworking of earthquake-
triggered landslide sediment by subsequent rainfall, i.e. Haiti (17). In all three cases, the long-term rainfall extremity was higher
than the short-term, which was neither extreme nor intense. We infer that the sequence of closely timed earthquakes and rainfall
triggers may matter less for triggering HTCs, as long as they follow extended rainy seasons.


Among the different topographical characteristics of the different AOIs, the sediment connectivity (IC) of hillslopes to the
drainage network did not show any clear relationship with any other terrain or rainfall variable, most likely due to its high
variability within each study area. Still, we consider that it would be worth further investigating the relationship between the
occurrence of HTCs and the IC of hillslopes on a more detailed scale, specifically by analyzing how the IC varies across the
terrain within each study area and how it relates to the runout of the HTCs.

In our database, around one-third of the AOIs are in Africa, and they stand out for their steep terrain and particularly coarse soils.
However, finding technical and academic information about them was difficult, as most references were limited to news reports
(Table A1). The high susceptibility of these regions and the scarcity of detailed data make the situation even more critical,
especially considering the recurrence of such events. For example, Jacobs et al. (2016) documented landslides and flash floods in
the Rwenzori Mountains, along the border between Uganda and the Democratic Republic of the Congo, from 1929 to 2014. They
found that debris-rich flash floods were the second deadliest events in the region, but also pointed out the data gaps in tropical
Africa, largely due to the coarse resolution of global datasets, which are often the only available sources of information. The
susceptibility of tropical Africa is expected to rise as a response to a warming climate: projections suggest that short-term rainfall
events in Eastern Africa could become more intense, potentially exceeding long-term rainfall accumulations (Palmer et al.,
2023). These findings should serve as a call to increase efforts to understand the occurrence of HTCs in tropical Africa.

Our cluster analysis reveals three groups of HTCs based on their terrain features, sediment composition, IC, and the intensity and
extremity of triggering rainfall. Cluster I includes events triggered by intense rainfall both in the short and long term, but
particularly in the short term, occurring on gentle slopes with predominantly sandy and silty soils. The susceptibility of these
areas may be linked to the presence of residual soils derived from granitic rocks. In China (10), Bai et al., (2021) described soils
derived from granite, consisting of residual soil overlying completely weathered granite. The upper layer had higher cohesion but
weakened when saturated, while the underlying granite was more permeable. These differences influenced infiltration, pore
pressure, and slope stability, with landslides often occurring at their interface. Similarly, in Brazil (2), Picanço et al., (2019)
found a transition in granite-derived soils from silty-clay topsoil to sandier horizons near bedrock, forming slip surfaces. In both
cases, abrupt changes at weathering boundaries played a key role in slope failure.

Several other studies also highlight the role of weathered granitic rocks in HTC formation. In Puerto Rico, landslides triggered
by Hurricane Maria were primarily in areas underlain by intrusive rocks such as (grano-)diorites (Bessette-Kirton et al., 2020).
In Colombia, the majority of the most destructive HTCs, locally called *avenidas torrenciales*, occurred in granitic headwater
basins (Aristizábal et al., 2020), and a very similar pattern was also noted in Malaysia (Legiman et al., 2023) and Japan (Chigira,
2001; Chigira et al., 2011; Hirata & Chigira, 2019). Additionally, the spheroidal weathering typical of granite rocks provides
large boulders, usually carried in the front of the flows, increasing their destructive power (Hirata & Chigira, 2019; Iverson,



1997). We infer that granitic terrains are particularly prone to HTC occurence under short-term intense rainfall, even in areas
with gentle slopes. However, this may be a particular trait of tropical regions with deeply weathered profiles, as de Haas et al.,
(2024) noted that, for temperate regions, quartz-rich igneous rocks tend to produce sandy and gravelly soils that are less prone to
debris flows compared to regoliths containing clay and silt, abundant in sedimentary and fine-textured metamorphic rocks.

The second cluster includes HTCs in terrain with steep slopes, high stream sediment connectivity, and a diverse range of soil
types, primarily silty to clayey. Triggering rainfall in this cluster was less extreme and intense compared to the other groups,
particularly in the short term. The lithologies were variable, including sedimentary rocks, recent volcanic deposits, andesites,
basalts, gneiss, and granites. This cluster also includes the three events triggered by both rainfall and earthquakes. In summary,
clusters 1 and 2 indicate that, for HTC to happen, gentler slopes demand higher rainfall and more channel connectivity, whereas
steeper, well-connected terrains require significantly less triggering rainfall.

Finally, Cluster 3 has two HTCs with exceptionally steep slopes and silt content, where the triggering rainfall was particularly
intense and extreme, especially in the long term. The combination of steep terrain, high silt content, and prolonged intense
rainfall over 60 to 90 days appears to have contributed to the susceptibility of these areas to HTCs.

## 6    Conclusions

The present study analyzes information about the interplay between rainfall, topographic and sediment characteristics in
triggering and propagating hillslope-torrential hazard cascades (HTCs) in tropical mountains. Rather than being driven by a
single factor, these events result from a combination of interacting drivers. We show that HTCs are not only the result of extreme
precipitation events, but more importantly, a consequence of both long- and short-term rainfall. While intense short-term
rainfalls can initiate HTCs, prolonged wet periods play a role in reducing soil stability and increasing runoff and sediment
mobilization. The majority of events were not triggered by the most extreme rainfall recorded, and we infer that these HTCs
occurred in supply-limited basins. The high sediment production rates in tropical environments may lead to shorter recurrence
intervals for HTCs compared to temperate regions.

Our analysis also recalls the challenge of capturing rainfall patterns in tropical regions. Differences between CHIRPS rainfall
estimates and local measurements indicate that coarse global datasets underestimate short-term rainfall variability, especially in
localized, intense rainfall episodes, which commonly occur in mountain regions. This limitation is particularly important in areas
where data are scarce and hazard assessments rely on global datasets.

The topographic setting of HTCs is very diverse. The role of sediment connectivity may deserve some future work looking into
the coupling of hillslopes to channels in more detail. We also find that many HTCs were tied to deeply weathered granite-derived
soils, where intense short-term rainfall following a prolonged rainy season destabilised the soil layer even on gentle slopes.

Cluster analysis revealed similarities between slope, rainfall intensity, and sediment granulometry: in regions with gentler slopes
and lower connectivity, HTCs require more extreme rainfall events to be triggered and sustained. In contrast, in steeper regions,
less extreme rainfall and less connectivity are sufficient for initiation. Beyond rainfall, earthquakes can also prepare and trigger
HTCs. Seismic activity can weaken slopes and make them more susceptible to failure when heavy rainfall occurs; trigger slope



failures where antecedent rainfall has saturated soils; and deposit excess sediment in channels to be mobilized by subsequent rainstorms.

470 Overall, our study highlights the importance of analyzing HTCs as a whole rather than as a combination of individual phenomena. This approach is essential for improving hazard assessment, eventually leading to risk reduction strategies in vulnerable tropical mountain regions.

## 7 Appendix A

**Table A1. Compilation of torrential hazard cascades in the tropics between 2009 and 2024**

| No | Date | Place | Lat, Long | Trigger | Area of Interest (km²) | Surveyed area (km²) | Land-slides | Area of land-slides (km²) | Deaths + Miss-ing | Trigger-ing rainfall (mm) | Time of rain-fall (hs) | Lithologies | Sources |
|---|---|---|---|---|---|---|---|---|---|---|---|---|---|
| 1 | 07-09/08/2009 | Taitung and Nantou County, Taiwan | 22.96, 120.84 | Rainfall (Thy-phoon) | 3827.40 | 7811 | 22705 | 274.16 | 699 | 2884 | 108 | Slates, Sandstones, Shales | C. W. Lin et al., (2011) |
| 2 | 11/03/2011 | Antonina, Paraná, Brazil | -25.59, -48.68 | Rainfall | 31.18 | 2.02 | 29 | 0.04 | 6 | 616.6 | 264 | Granite | Dias, McDougall, et al., (2022); Picanço & Nunes, (2013) |
| 3 | 18/05/2015 | Salgar, Antio-quia, Colombia | 5.97, -76.04 | Rainfall | 66.81 | - | 40 | - | 104 | 160 | 20 | Monzogranites | Marin et al., (2021); Ruiz-Vásquez & Aristizábal, (2018) |
| 4 | 25/11/2016 | Bijagua, Ala-juela, Costa Rica | 10.73, -85.11 | Earth-quake and rainfall (Hurri-cane) | 38.70 | 27.00 | 942 | - | 8 | 291 | 24 | Recent volcanic deposits | Quesada-Román et al., (2019) |
| 5 | 31/03/2017 | Mocoa, Putu-mayo, Colombia | 1.16, -76.66 | Rainfall | 14.47 | 53.40 | 629 | 1.2 | 333 | 130 | 3 | Monzogranite, Conglome-rates, Sands-tones, Silsto-nes | Prada-Sarmiento et al., (2019) |
| 6 | 19/09/2017 | Saint George Parish, Domini-ca | 15.30, -61.34 | Rainfall (Hurri-cane) | 22.73 | 591.95 | 9960 | 10.3 | 64 | 579 | 23 | Basaltic lavas, tuff, ash, limestones, conglomer-ates | Brookes & Gabriel, (2018); Government of the Commonwealth of Dominica, (2017); Taylor et al., (2023) |
| 7 | 07/07/2018 | Kure, Hiroshi-ma, Japan | 34.30, 132.68 | Rainfall (Thy-phoon) | 31.81 | 1907 | 2934 | - | 116 | 368 | 47 | Granites, Dacites, Tuffs, Conglomer-ates | Geospatial Information Authority of Japan, (2018); Hashimoto et al., (2020); Rodrigues Neto et al., (2023); Rodrigues Neto & Bhandary, (2023) |
| 8 | 17/08/2018 | Kodagu, Karna-taka, India | 12.46, 75.74 | Rainfall | 45.32 | 4102 | 343 | - | 20 | 768 | 36 | Gneiss, Granulite, Khondalites, Granites | Dayananda, (2019); Meena et al., (2021) |
| 9 | 17/03/2019 | Chimanimani, Zimbabwe | -19.87, 32.83 | Rainfall (Cyclone) | 81.93 | 2375 | 385 | 12.1 | 625 | 400 | 120 | Quartzite, Diorite, limestone | Muchaka et al., (2022) |



| No | Date | Location | Coordinates | Trigger | | | | | | | | Lithology | References |
|---|---|---|---|---|---|---|---|---|---|---|---|---|---|
| 10 | 16/03/2019 | Sentani, Papua, Indonesia | -2.53, 140.51 | Rainfall | 72.74 | - | - | - | 183 | - | - | Ultramafic volcanic rocks, Meth-amorphic rocks | Gasica et al., (2020); Kristiawan et al., (2020) |
| 11 | 10-13/06/2019 | Mibei, Guang-dong, China | 24.64, 115.3 | Rainfall | 21.47 | 14.76 | 327 | 0.75 | 0 | 281.3 | 96 | Granite | Bai et al., (2021) |
| 12 | 09/08/2019 | Wayanad, Malappuram and Kozhikode, Kerala, India | 11.31, 76.48 | Rainfall | 110.73 | 1258 | 756 | 5.34 | 120 | 440 | 144 | Charnockite, granulites, gneiss | Jain et al., (2021); Wadhawan et al., (2020) |
| 13 | 24/11/2019 | Tamkal, West Pokot, Kenya | 1.37, 35.46 | Rainfall | 52.69 | - | 2319 | 3.8 | 120 | 400 | 72 | - | Scheip & Wegmann, (2021); Schlögel et al., (2020) |
| 14 | 04/12/2019 | Nyempundu, Cibitoke, Burundi | -2.62, 29.09 | Rainfall | 16.50 | - | 318 | - | 37 | - | - | - | Deijns et al., (2022) |
| 15 | 18/04/2020 | Kipchumwa and Chesegon, West Pokot, Kenya | 1.28, 35.61 | Rainfall | 21.61 | - | - | - | 29 | - | - | Gneiss, Phonolites | Loice et al., (2021) |
| 16 | 17-18/12/2020 | Presidente Getúlio, Santa Catarina, Brazil | -27.09, -49.61 | Rainfall | 27.04 | - | 143 | 0.54 | 21 | 122 | 120 | - | Lucchese et al., (2022); Schramm & Osako, (2023) |
| 17 | 16/08/2021 | Pic Macaya National Park, Haiti | 18.38, -74.02 | Earth-quake and Rainfall (Cyclone) | 240.37 | 2700.00 | 8444 | 45.8 | - | 254 | 24 | Limestone, Chert, Basalt | Martinez et al., (2021); Zhao et al., (2022) |
| 18 | 25/02/2022 | Mount Tala-kmau, West Sumatra, Indonesia | 0.09, 99.99 | Earth-quake and Rainfall | 33.19 | 150.00 | - | 6.00 | - | - | - | Andesite, Basalt | Fang et al., (2024); UNITAR & UNOSAR, (2022) |
| 19 | 13/03/2023 | Chiradzulu, Malawi | -15.69, 35.16 | Rainfall (Cyclone) | 33.19 | - | - | - | - | 1078 | 96 | - | Government of Malawi, (2023) |
| 20 | 04/05/2023 | Bushushu, Nyamukubi, Luzira and Chabondo, South Kivu, DRC | -2.019, 28.905 | Rainfall | 13.44 | 90.00 | - | 0.9 | 5438 | - | - | - | International Disasters Charter, (2023); MapAfrica - African Development Bank Group, (2023) |
| 21 | 03/12/2023 | Mount Hanang, Man-yara,Tanzania | -4.44, 35.40 | Rainfall | 11.60 | - | - | - | 89 | 102 | 24 | Granites, volcanic aglomerates, Tuffs | Mulaya et al., (2024) |
| 22 | 29/04/2024 | Mai Mahiu, Nakuru, Kenya | -0.933, 36.616 | Rainfall | 6.04 | - | - | 2 | 132 | - | - | - | Reuters, (2024); UNITAR & UNOSAT, (2024) |

475



## 8 Code and data availability

The ALOS PALSAR Digital Elevation Model is freely available. CHIRPS rainfall estimates are available in https://www.chc.ucsb.edu/data/chirps. SoilGrids global soil information is available in https://soilgrids.org/.

## 9 Author contributions

MIAC, OK, MH and EA conceptualized this study. MIAC carried out the analysis, PV carried out the rainfall extremity analysis. MIAC and OK led the writing, with contributions from EA, MH and PV.

## 10 Competing interests

The authors declare that they have no conflict of interest.

## 11 Financial support

MIA was funded through the BMBF project 403 BB-KI Chips (grant 16DHBKI020). PV was funded by the Deutsche Forschungsgemeinschaft (grant no. GRK 2043, project number 251036843). EA was funded by a Georg Forster Research Fellowship for Experienced Researchers from the Alexander von Humboldt Foundation.

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
