# Peer review of "Hillslope-Torrential Hazard Cascades in Tropical Mountains"

_EGUsphere, 2025_

## Author Response (AR1)

**Response to Reviewer #1**

Dear reviewer,

We sincerely thank you for the comments on our manuscript. We are pleased to hear that you found the topic as relevant as we do. Below we provide a point-by-point response to your suggestions (text additions in the revised manuscript are shown underscored).

**Comment 1:** *"At the end of the abstract, state the applicability of the study, i.e., to which institutions and organizations the obtained results will be useful."*

**Response:** We have revised the final sentences of the abstract to briefly state the applicability of our findings. We have added the following sentence: "Our findings highlight the need to prioritize hazard assessment in tropical mountains and may support the work of researchers and disaster risk agencies towards early warning and land-use planning in underreported regions like Africa".

**Comment 2:** *"At the beginning of the introduction, a major event is described, a natural disaster that caused great material and human losses."*

**Response:** Thank you for acknowledging this. No changes were necessary.

**Comment 3:** *"At the end of the introduction, you can add a few sentences about the following titles and subtitles."*

**Response:** To improve clarity and guide the reader through the structure of the manuscript, we have added a brief overview of the main sections and subsections at the end of the introduction, as recommended.

The revised final paragraph of the introduction now states: "To address some of the previously mentioned shortcomings and contribute to the knowledge base of hazard cascades of coupled hillslope and torrential processes in the Tropics, our goal is to estimate the relative roles of topography characteristics, soil granulometry, sediment connectivity, and the extremeness of antecedent and triggering rainfall. We begin in Section 2 with a review of current knowledge on hillslope torrential cascades, focusing particularly on the conditions that favor their occurrence in the Tropics. We aim to classify these events based on topographic, granulometric, and rainfall-related variables to identify patterns and distinguish event types. Section 3 describes the methodology and data used for the classification analysis, with the findings presented in Section 4, further discussed in Section 5, and summarized in Section 6."

**Comment 4**: *"Subtitle 2.1. In Figure 1, you can add coordinates to fully show the geographical location."*

**Response:** Since Figure 1 presents a 3D orthographic view created in QGIS, it is not possible to automatically display coordinate gridlines as in a 2D map. However, to address your comment, we have added the geographic coordinates of the central point of the image and clarified this limitation in the figure caption.

We thank you again for your constructive feedback, which has helped us improve the clarity and usefulness of our manuscript.

Sincerely,
The Authors

**Response to Reviewer #2**

Dear reviewer,

We thank the reviewer for their useful comments. We appreciate the recognition of our work and the suggestions for improving the manuscript. Below we provide detailed responses to each of the points raised and the text added in the revised manuscript, which are shown underscored.

**Comment 1:** *"The reviewer suggests the authors to give a look to the classification of torrential hazard provided by Brenna et al. (2020), Church and Jacob (2020), and Laigle and Bardou (2022)."*

**Response:** Thank you for this suggestion. We have reviewed the references and included them in Section 2.1, *Hillslope–Torrential Cascades*. The works by Brenna et al. (2020) and Church and Jacob (2020) did not bring additions or conceptual changes to the manuscript but were cited to reinforce some of the definitions. In particular, the paper by Laigle and Bardou (2022) was a very interesting finding for us, given how it frames mass movement processes in mountainous regions as cascades. We found this approach especially interesting for our study, and we included this concept into the manuscript. The text now reads:

"Hungr et al. (2014) use the term "debris flow" in a broad sense to encompass a cascading sequence of geomorphic processes that initiate on hillslopes, such as shallow landslides and debris avalanches, and subsequently propagate along drainage channels as debris flows and floods. This integrated definition emphasizes the slope-to-channel sediment transfer and the coupling between hillslope and fluvial processes at the basin scale as a cascading system of sediment transfer in steep mountain catchments. This cascade is not limited to single event but involves a sequence of interactions, reinforcing the importance of analyzing torrential hazards at the scale of the catchment system (Laigle & Bardou, 2022)"

**Comment 2:** *About the debris flows that origin for entrainment of deposits from landslide or cliff failure into runoff, these are the case of Manival torrent in France (Theule, 2012) and Ru Secco torrent in Italy (Barbini et al., 2024). Barbini et al. (2024) also provides a schematic view of debris-flow triggering where debris flow can form when runoff entrain sediments from the deposits given by bank, slope and cliff failures.*

**Response:** Thank you for sharing these two case studies. While we believe that these examples do not change the main concept presented in our manuscript, they provide interesting cases that support our discussion.

The study by Theule (2012) offers detailed documentation of sediment dynamics in supply-limited basins in temperate environments. We found this helpful and have incorporated a reference to it in our discussion section to further support our argument regarding sediment availability and its role in the initiation of THC. The full sentence now reads:

"The frequency of torrential events in supply-limited basins depends on the specific sediment recharge rates, producing less frequent but more extreme events than transport-limited basins do, where unlimited sediment supply means torrential events are triggered more by critical rainfall (Jakob & Hungr, 2005). Some observations in temperate mountainous catchments highlight the seasonal nature of cycles of sediment recharge, storage and flushing, where torrential hazards are triggered by critical discharge, capturing sediment from channel storage zones (Theule et al., 2012). While this recharge–flushing dynamic is established in temperate mountain settings, comparable data are scarce in tropical environments, where the timing and magnitude of sediment recharge cycles remain under-characterized."

Regarding Barbini et al. (2024), while we recognize the value of their review of sediment entrainment in debris-flow triggering, we ultimately decided not to include the reference. The reasons for it are that the focus of the paper is primarily on the design of sediment control structures, and that the event descriptions do not introduce new conceptual elements than those already discussed in the manuscript.

**Comment 3:** *Lines 94-96 "The resulting flow can be a debris flow or debris flood, or it can alternate between the two, for example when a channelized debris flow outpaces slower-moving flood waves of lesser sediment content; by bed erosion that causes flow bulking; or by an increase in discharge and sediment deposition of the debris flow" It could occur also an hyperconcentrated flow depending on the size of sediments and bed slope (Mark, 2017; Brenna et al., 2020; Laigle and Bardoux, 2022).*

**Response:** Thank you for this observation. We have changed the sentence to include the possibility of hyperconcentrated flows, citing Laigle and Bardou (2022), who explicitly refer to this case. The final sentence reads:

"The resulting flow can be a debris flow, debris flood, or hyperconcentrated flow, or it can alternate between them, for example when a channelized debris flow outpaces slower-moving flood waves of lesser sediment content; by bed erosion that causes flow bulking; by an increase in discharge and sediment deposition of the debris flow (Church & Jakob, 2020); flow blockages caused by woody debris that briefly increase flow momentum, or by a mudflow entering the drainage network (Laigle & Bardou, 2022).

**Comment 4**: *"Line 200 - authors should justify the choice of 90 days for the computing of the antecedent rainfall."*

**Response:** We now include a justification for the 90-day antecedent rainfall period selection in Section 3.2.2 *Rainfall analysis*. The paragraph now reads:

"Antecedent rainfall is defined as the total rainfall accumulated over the 90 days preceding the event. This 90-day period captures seasonal-scale rainfall accumulation in tropical regions associated with monsoonal or ITCZ dynamics. Furthermore, many authors consider the 90-day window to be a representative and widely used measure of antecedent rainfall in landslide hazard (Cardinali et al., 2006; Zêzere et al., 2005)."

**Comment 5:** "*Line 280 - rainfall is estimated in mm: it should be mm/h or mm/day.*"

**Response:** Thank you for this observation. We agree that rainfall intensity should be expressed in mm/h or mm/day. However, in this case, we are referring to the total accumulated rainfall over specific periods (1 day, 3 days, and 90 days), which is expressed in millimeters (mm). We have revised the manuscript to clarify this by specifying that the values refer to accumulated rainfall, and have added the corresponding time periods to avoid confusion. The paragraph reads now:

"Rainfall characteristics are analyzed in terms of intensity (Figure 8) and extremity (Figure 9 and Figure 10). Figure 8 shows the relationship between 1-day and 3-day with the 90-day rainfall accumulation for each AOI. The triggering rainfall intensities varied between 10 and 100 mm accumulated on the day of the event, and 90-day accumulated totals from 700 to 1300 mm. The lowest triggering and antecedent rainfall for a HTC was associated with an event in Brazil (16) with only 11 mm on the event day, and 370 mm accumulated over 90 days before."

**Comment 6:** *"Line 290 - This result is very interesting."*

**Response:** Indeed, we agree that this is an interesting result. As noted in the manuscript, CHIRPS has limitations in capturing localized and short-duration rainfall, but the pattern still shows an important relationship between triggering and antecedent rainfall in the initiation of HTC events.

**Comment 7:** *Lines 365-367 "However, CHIRPS estimated 25 mm on the event day. Similarly, for the 2020 Brazil event (16), CHIRPS estimated 11 mm on the event day, while Michel et al., (2021) reported 124 mm in a single day based on local rain gauge data. These inconsistencies were evident throughout the database, with absolute differences between reported and actual values decreasing as the accumulation period increased." Please note that Bernard and Gregoretti (2021) observed that changing the gauge used for the calibration, the radar estimates could highly vary.*

**Response:** Thank you for this observation and for referring us to the study by Bernard and Gregoretti (2021). While their work focuses on the use of rain gauges to correct high-resolution radar reflectivity data (0.5 × 0.5 km²), our dataset uses CHIRPS, a satellite-based estimate that combines infrared satellite data with rain gauge information of a much coarser resolution (~5 × 5 km²). While both radar and CHIRPS estimates have biases, especially in mountainous regions, the scale and nature of their uncertainties are probably different. Nonetheless, the main point remains valid: remote precipitation estimates underestimate localized, high-intensity rainfall. For this reason, and given the methodological and scale differences of the study of Bernard and Gregoretti (2021), we decided not to include this reference in the manuscript, although we recognize its importance for early warning systems based on radar data.

**Comment 8:** *"The references Kaitna et al. (2024)a,b are the same."*

**Response:** Thank you for pointing this out. We have removed the duplicate reference.

We are grateful for the reviewer's thoughtful recommendations, many of which helped us improve the manuscript with additional details. Thank you also for identifying small errors and inconsistencies that have now been addressed.

Sincerely,

The Authors